

# Investigating the impact of HARMONIE-WINS50 (cy43) and LOTOS-EUROS (v2.2.002) coupling on $NO_2$ concentrations in The Netherlands

Andrés Yarce Botero [1,2], Michiel van Weele [3], Arjo Segers [4], Pier Siebesma [2,3], and Henk Eskes [3]

[1]Mathematical Physics, Delft University of Technology, Mekelweg 4, 2628 CD Delft EWI TuDelft, the Netherlands
[2]Department of Geoscience & Remote Sensing, Delft University of Technology, Stevinweg 1, 2628CN, CITG TuDelft, the Netherlands
[3]Royal Netherlands Meteorological Institute (KNMI), De Bilt, 3730 AE, the Netherlands
[4]Air quality & Emissions Research TNO, Princetonlaan 6, 3584 CB Utrecht, the Netherlands

**Correspondence:** Andrés Yarce Botero (a.yarcebotero@tudelft.nl)

**Abstract.** Meteorological fields calculated by Numerical Weather Prediction (NWP) Models drive offline Chemical Transport Models (CTM) to solve the transport, chemical reactions, and atmospheric interaction over the geographical domain of interest. In this way, forecasts and (re-)analyses provided by NWP can be used for air quality forecasting, climate modeling, and environmental studies. The more precise the meteorological input data represents the atmospheric dynamics, the better

the CTM represents pollutant transport, mixing, and the subsequent impact on surface air quality. HARMONIE (HIRLAM ALADIN Research on Mesoscale Operational NWP in Euromed) is a state-of-the-art non-hydrostatic NWP community model used at several European weather agencies to forecast weather at the local and/or regional scale. In this work, the HARMONIE WINS50 (cycle 43 cy43) reanalysis data set at a resolution of $0.025° × 0.025°$ covering an area surrounding the North Sea for the years 2019-2021 was offline coupled to the state-of-the-art model LOTOS-EUROS (v2.2.002), which is a CTM that

is one of the members of the Copernicus Atmosphere Monitoring Service (CAMS), an ensemble of CTMs that is used to produce operational air quality forecasts over Europe and at a higher resolution also over the Netherlands. The impact on simulated $NO_2$ concentrations of using meteorological fields from HARMONIE in LOTOS-EUROS compared to the use of fields from ECMWF (here used at $0.7° × 0.7°$) is evaluated against ground-level sensors and TROPOMI tropospheric $NO_2$ vertical columns. Furthermore, the difference between crucial meteorological input parameters such as the boundary layer height

and the vertical diffusion coefficient between the hydrostatic (ECMWF) and non-hydrostatic (HARMONIE) model fields is studied, and the vertical profiles of temperature, humidity, and wind are evaluated against meteorological vertical profile observations at Cabauw in The Netherlands. The results of these first evaluations of the LOTOS-EUROS model performance in both configurations are used to investigate current uncertainties in air quality forecasting in relation to driving meteorological parameters and to assess the potential for improvements in high-resolution air quality forecasting episodes based on the HAR-

MONIE NWP model.

Keywords: WINS50, LOTOS-EUROS, HARMONIE (cy43), ECMWF, Offline coupling, TROPOMI



# 1 Introduction

Meteorological fields calculated by Numerical Weather Prediction Models (NWP) provide necessary input to Chemical Transport Models (CTM) to solve the emission, transport, chemical reactions, and other atmospheric interactions of pollutants over the spatiotemporal domain of interest (Chang, 1980; El-Harbawi, 2013; Khan and Hassan, 2020). Meteorological parameters related to transport and mixing have a direct impact the surface air quality simulated by the CTM. A NWP model with a higher spatial resolution and better capabilities for resolving boundary layer turbulence dynamics and convective processes would provide the CTM with more accurate input parameters to predict the movement of pollutants, especially in the lowest kilometer(s) of the troposphere (Pielke and Uliasz, 1998).

However, it is important to note that the spatial resolution of the NWP model is not the only factor. Other factors may include the model's ability to accurately represent small-scale phenomena, turbulence dynamics, and convective processes (non-hydrostatic), compared to models that replace the vertical momentum equation by hydrostatic equilibrium (SAITO et al., 2007). Also, the quality of (operational) meteorological input is constantly improved through the data assimilation applied in NWP (Marseille and Stoffelen, 2017; Bengtsson et al., 2017; Lorenc and Jardak, 2018) which can reduce the model uncertainty representation. Overall, it is important to carefully consider the uncertainty of the meteorological driving parameters in a CTM, as these parameters can significantly affect the accuracy and reliability of the simulated air quality predictions.

HARMONIE (HIRLAM ALADIN Research on Mesoscale Operational NWP in Euromed), (Bengtsson et al., 2017) is the operational high-resolution NWP model that is used in The Netherlands (Haakenstad et al., 2021). The WINS50 is the dataset that is used in this work, it is an homogeneous HARMONIE reanalysis focusing on the North Sea region, developed by a consortium of Whiffle, TU Delft, and KNMI. The dataset covers the years 2019 to 2021 and has been created using HARMONIE cycle 43. It was evaluated for one year by (van Stratum et al., 2022) to show how and to what extent current wind farm structures in the north sea can cause effects on the meteorology at local to regional scales (Verzijlbergh, 2021; Kalverla et al., 2019; Baas et al., 2022)

LOTOS-EUROS (LOng Term Ozone Simulation-EURopean Operational Smog model) is a chemical transport model that simulates the formation and transport of pollutants and trace gases in the atmosphere (Manders et al., 2017). The processes in the model include emission, advective transport, turbulent mixing, chemical reactions, wet- and dry deposition, and sedimentation. In most applications, the model is driven by meteorological input from ECMWF, but in this study, it has been coupled with the HARMONIE NWP to provide a more comprehensive understanding of the formation and transport of air pollutants in the BeNeLux countries and North Sea region. In earlier studies, other meteorological drivers have been offline one-way directional coupled to the LOTOS-EUROS model, including WRF (Escudero et al., 2019), COSMO (Thürkow et al., 2021), and, in RACMO (Manders-Groot et al., 2011)a two-way coupling was implemented with frequent coupling between NWP and air quality simulations to provide insight in the impact of meteorological conditions on air pollutants, and vice versa the impact of trace gasses and aerosol on weather and climate via for example the radiation budget.



In a previous study by (Ding, 2013), the impact of using HARMONIE (cy36) as a meteorological driver for LOTOS-EUROS (v1.8) was compared with using European Centre for Medium-Range Weather Forecasts (ECMWF) meteorology. That study found large differences in the meteorological variables obtained from the two drivers, especially at the coast, over forest regions, and in urban areas. However, the surface temperature, relative humidity, and wind patterns were found to be very similar between the models. Since this previous study, various updates and improvements have been made to both the HARMONIE NWP model and the LOTOS-EUROS CTM, which have involved into cycle 43 and v2.2002 respectively. Therefore, conducting a new assessment and reassessing their coupled performance is valuable.

Section 2 of this paper introduces the methodology used in the study. It includes a description of the two meteorological input fields with the configurations made for the coupling with the version of LOTOS-EUROS used in this study. The coupling procedure between the meteorological driver and the CTM is explained in this section, along with the list of variables taken into account and any necessary calculations or assumptions for their correct ingestion by the CTM. Section 3 presents the results of the model simulations and their evaluation against ground-base observations and satellite-observed trace gas plumes. The comparison with observations is important to better assess the differences between the model simulations. The paper's final section, Section 4, discusses our results and provides the conclusions on the coupling of HARMONIE WINS50 NWP to LOTOS-EUROS as drawn from this study. Additionally, the potential for improvements in high-resolution air quality forecast offline driven by high-resolution non-hydrostatic meteorological parameter fields is assessed.

## 2    Methodology: Coupling of Meteorological Drivers to the Chemical Transport model

### 2.1    LOTOS-EUROS driven by ECMWF meteorology

LOTOS-EUROS is a large-scale three-dimensional CTM that simulates air pollution in the lower troposphere by solving a differential equation involving different operators, such as the transport operator, the chemical reaction operator, and the emissions/deposition operator. This operators are executed sequentially on a 3D set of grid cells covering the troposphere over the domain of interest. The horizontal advection is driven by horizontal winds (U, V) that are part of the meteorological input. When driven by ECMWF meteorology, the model calculates the vertical wind component (W) through the convergence and divergence of the horizontal winds. Turbulence driven vertical diffusion is modelled with a seperate operator. The chemistry operator simulates the chemical production and loss terms from the different chemical reactions in the atmosphere. A Carbon Bond Mechanism with 81 reactions (Schaap et al., 2008) is used to describe the gas-phase chemistry, and interaction with aerosols follows the ISORROPIA parameterization (Fountoukis and Nenes, 2007). The dry deposition operator is parameterized following the resistance approach (Wichink Kruit et al., 2012). The wet deposition operator includes the below-cloud scavenging for gases (Schaap et al., 2004).

LOTOS-EUROS receives the ECMWF meteorological fields on a regular longitude-latitude grid, which is then interpolated to the target grid that is either regular longitude-latitude too or uses a different projection. The vertical layers of the model are defined as a coarsening of the ECMWF hybrid sigma-pressure layers. The meteorological fields received from the ECMWF data include 3D fields of pressure, wind vectors, temperature, and humidity, as well as 2D fields of mixing layer height,





precipitation rates, cloud cover, and other boundary layer and surface variables, among others, listed in table 1 in the following

section, are used to drive the transport and concentration rates of pollutants in the atmosphere.

## 2.2  LOTOS-EUROS driven by HARMONIE meteorology

The HARMONIE (HIRLAM ALADIN Research on Mesoscale Operational NWP in Euromed) is a non-hydrostatic convection-permitting Numerical Weather Prediction model (Engdahl et al., 2020; Clark et al., 2016). In a non-hydrostatic model, the vertical momentum equation is solved directly instead of applying the hydrostatic approximation, which frequently fails dur-

ing extreme weather events (Gibbon and Holm, 2011). HARMONIE incorporates various dedicated sub-models to describe atmospheric processes. One of these models is SURFEX, which simulates processes such as temperature and water balance, radiation balance, and heat transport at the surface and in the soil (Viana Jiménez and Díez Muyo, 2019). The model accounts for various types of land surfaces and processes at and below the surface to describe the interaction between the atmosphere and the surface.

Similar as the ECMWF model, the HARMONIE model uses terrain-following hybrid sigma-pressure layers that are defined by surface pressure and hybrid level coefficients provided in the data files; Although the HARMONIE model could provide non-hydrostatic vertical advective fluxes, it was decided to perform a coupling with HARMONIE based on the same approach as used for ECMWF variables (see our discussion in Section 4).

The particular HARMONIE simulation for this project comes from the "WINS50" project. TUDelft, Whiffle, and KNMI

have formulated the WINS50 project in the framework of the TKI Wind op Zee R&D 2019 ( www.wins50.nl ). The WINS50 model was run for 2019-2021 to produce winds undisturbed by wake effects (extension of the Dutch Offshore Wind Atlas DOWA) and disturbed winds (wake-DOWA). The simulation was performed with the LOTOS-EUROS driven with ECMWF meteorology (EC_LE) and the LOTOS-EUROS driven with the HARMONIE meteorology (HA_LE ). One recent comparison of the HARMONIE model for the North Sea with other models and also observation from a mast to compare a couple of

vertical levels can be found in (Kalverla et al., 2019)

First, the data was moved from ECGATE to SNELLIUS. LOTOS-EUROS ingested the variables selected from the HARMONIE WINS50 correspondent to the ECMWF variables based on the coupling choices specified in the following section. Second, the decision about whether direct or indirect mapping should be done and what to do with missing variables is taken. Third the labeling and timestamp frequency and time bounds were corrected and the direct paths to find the data and me-

teorological files were generated for the LOTOS-EUROS files. Mapping Halflevel altitudes with Half level pressures with coefficients calculation was done using specific routines generated that additionally flip the order of some needed variables. Additionally, determining and converting the variables needed in accumulated or instantaneous formats was another task that was paid attention to.

### 2.2.1  Coupling choices

To ensure successful coupling in the system (HA_LE ), a systematic approach was taken comparing the available ECMWF and HARMONIE fields. This involved classifying the variables into three categories: static, surface, and 3D fields in Table



1. The table was created to compare the variables' acronyms, units, and availability between the two systems. The resulting comparison helped identify which variables could be used immediately, which required further calculations, and which needed to be excluded due to unavailability. The coupling strategy was built under the assumption we wanted to emulate how currently,

the LOTOS-EUROS ingest data from the ECMWF fields (EC_LE). This table represents the static variables in purple, the dynamical two-dimensional in red, and the dynamical three-dimensional fields in green. This thorough approach ensured that the (HA_LE ) system is technically coupled, allowing for the generation of accurate and comprehensive CTM fields driven by this new source of meteorology information.

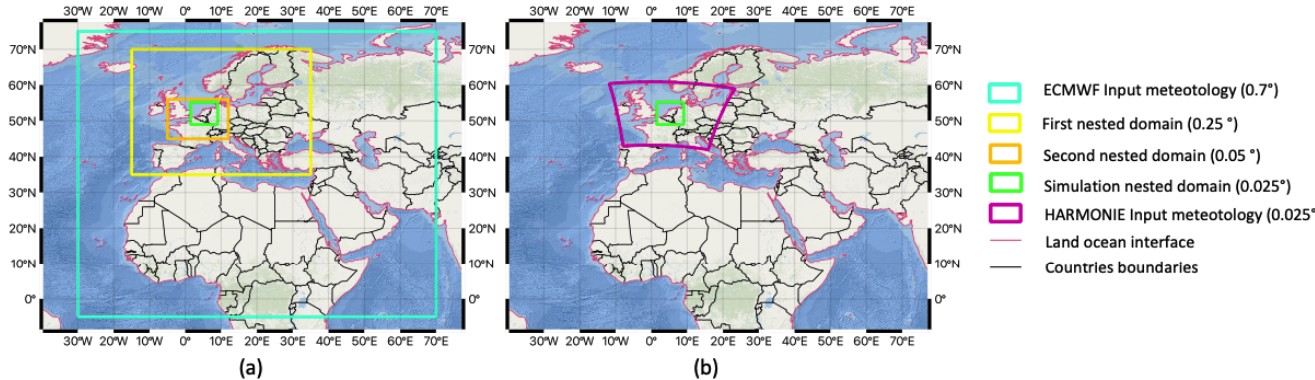

**Figure 1.** Configurations of the two meteorology drivers for the LOTOS-EUROS CTM. On the left LOTOS-EUROS nested domains using ECMWF meteorology, and on the right the LOTOS-EUROS domain using HARMONIE meteorology. Both configurations use boundary conditions from CAMS. (map from Natural Earth collection (https://www.naturalearthdata.com/ 1:50m Natural Earth I with Shaded Relief and Water )

Table (2) shows the LOTOS-EUROS configuration settings for the simulations performed in this study. Those configuration

settings are essential for understanding the methods used in this study and for interpreting the results, with the main difference between the system from the meteorology input. The other parameters were kept equal to isolate the effects of the meteorology changes and attribute any discrepancies to this factor. Using different meteorological models allows for comparing the resulting $NO_2$ concentrations while keeping the other parameters constant, allowing for a more accurate assessment of the effects of the meteorology changes on the simulations. The table lists the different parameters used in the two LOTOS-EUROS config-

urations, including the meteorological data source, the chemical boundary conditions, the emissions, land use, the horizontal resolution for the objective domain and the nested domains, and the time step used for the simulations.

### 2.2.2 About the computational aspects

The Figure (1) shows two spatial configurations of the LOTOS-EUROS CTM that use different meteorology drivers. The configuration on the left has three nested domains and uses ECMWF meteorology, while the configuration on the right has





**Table 1.** Comparison between the ECMWF (fields from ERA5 Levels 137 converted to levels 42) and the HARMONIE WINS 50 (cy 43) variables, their acronyms, and units. The variables are divided into static (purple), dynamical two (red), and three dimensions (green). Variables with the symbol (*) were converted from instantaneous to accumulated. The variables underlined were calculated with other available variables

| ECMWF | | HARMONIE | | Units |
|---|---|---|---|---|
| Acronym | Long name | Acronym | Long name | |
| **1- Static surface fields** | | | | |
| lsm | Land sea mask | lsm | Sea area fraction | [0,1] |
| orog | Orography | orog | Surface altitude | [m] |
| slt | Soil type | slt | Soil type | |
| **2- Surface and other dynamic 2D model** | | | | |
| blh | Boundary layer height | zmla | Atmosphere boundary layer thickness | [m] |
| tsurf | Surface temperature | ts | Surface temperature | [K] |
| dsurf | Surface dewpoint | | alculated from hhus and ts using stuhl approximation | [K] |
| u10 | 10 meter wind vector | uas | Eastward Near-Surface Wind Velocity | [m s$^{-1}$] |
| v10 | 10 meter wind vector | vas | Northward Near-Surface Wind Velocity | [m/s] |
| sd | Snowdepth | snw | Surface snow amount | [m] |
| sstk | Sea surface temperature | sst | Sea surface temperature | [K] |
| swvl1 | Volumetric soil water layer N | wsa_L01.P01 | Volume Fraction Of Liquid Water In Soil Layer 1 | [m$^3$ m$^{-3}$] |
| swvl2 | Volumetric soil water layer N | wsa_L02.P02 | Volume Fraction Of Liquid Water In Soil Layer 2 | [m$^3$ m$^{-3}$] |
| swvl3 | Volumetric soil water layer N | wsa_L03.P03 | Volume Fraction Of Liquid Water In Soil Layer 3 | [m$^3$ m$^{-3}$] |
| swvl4 | Volumetric soil water layer N | wsa_L04.P04 | Volume Fraction Of Liquid Water In Soil Layer 4 | [m$^3$ m$^{-3}$] |
| tcc | Total cloud coverage | clt | Total cloud fraction | [0 1] |
| zust | Friction velocity grass | | alculated with square(Tauu+Tauv)/density | |
| sshf | Surface sensible heat flux | hfss | Accumulated Surface Upward Sensible Heat Flux | [J m$^{-2}$] |
| slhf | Surface latent heat flux | hfls_eva | Accumulated Upward latent flux of evaporation (*) | [J m$^{-2}$] |
| cp | Convective precipitation | prrain | Accumulated rain (*) | [kg m$^{-2}$] |
| lsp | Large scale precipitation | prrain | Accumulated rain (*) | [kg m$^{-2}$] |
| sf | Snowfall | prsn | Snowfall amount (*) | [kg m$^{-2}$] |
| ssrd | Surface solar radiation downwards | rsds | Accumulated Surface Downwelling Shortwave Radiation (*) | [J m$^{-2}$] |
| sp | Surface pressure | ps | Surface air pressure | [Pa] |
| **3- Dynamic model 3D fields** | | | | |
| hp | Half level pressure | ps | alculated from the half level coefficients and surface pressure | [Pa] |
| t | Temperature | ta | Air temperature | [K] |
| q | Specific humidity | hus | Specific humidity | [kg kg$^{-1}$] |
| v | v component of wind | va | Northward wind velocity | [m s$^{-1}$] |
| u | u component of wind | ua | Eastward wind velocity | [m s$^{-1}$] |
| cc | Cloud cover | clt | Total Cloud Fraction | [0-1] [kg kg$^{-1}$] |
| clwc | Specific cloud liquid water content | clw | Cloud water | [kg kg$^{-1}$] |



**Table 2.** LOTOS-EUROS configuration settings for the simulations in this work. The principal difference is the input of the meteorology. The rest of the parameters were not touched to attribute the discrepancies only to the change in meteorology. coordinates of the domain presented in [Lat N, Lon E]

| Simulation periods | 1 April to 30 April 2019 |
|---|---|
| Meteorology | ECMWF; Temp.res: 1h; Spat.res: 0.7° |
| Meteorology | HARMONIE WINS50; Temp.res: 1h; Spat.res: 0.025° |
| Initial and boundary conditions | CAMS (D1). Temp.res: 1h. Spat.Res: 0.9° |
| Anthropogenic emissions | CAMS Spat.res: 0.1° |
| Biogenic emissions | MEGAN Spat.res: 0.1° |
| Fire emissions | MACC/CAMS GFAS Spat.res: 0.1° |
| Land use | CLC 2012. Spat.res: 0.01° |
| Topography | GMTED2010. Spat.res: 0.002° |
| HARMONIE WINS50 (Lagrangian projection) | [-8.5°, 43°] x [16°, 42°]x[23°, 59°] x [-12°, 61°] |
| ECMWF [Lat N x Lon E] | [-5°, 75°] x [-30°, 70°] |
| First ECMWF nested domain [Lat] x [Lon] | [35°, 70°] x [-15°, -35°] |
| Second ECMWF nested domain[Lat] x [Lon] | [45°, 18°] x [5°, -60°] |
| Objective simulation grid [Lat] x [Lon] (Both configurations) | [49°, 13.27°] x [1.5°, -65.94°] |

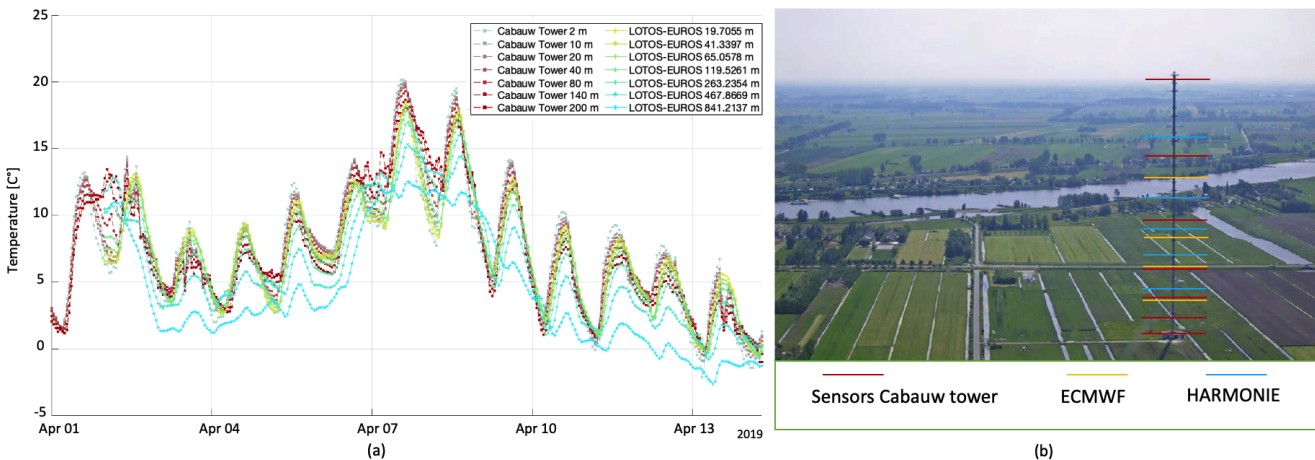

**Figure 2.** (a) Time series of the temperature from the ECMWF meteorology compared with the Cabauw observations compared for different levels and (b) the image from the Cabauw tower (lat 51.96° N, lon 4.89°W) with three colors for the sensors,the ECMWF and HARMONIE model levels for comparison, aerial photo image modified from (Apituley et al., 2008)



one domain and uses HARMONIE meteorology. Both configurations use boundary conditions from CAMS. Using nested domains in the first configuration allows for more precise modeling of atmospheric conditions in areas with coarse boundary information. In contrast, the second configuration has a high-resolution meteorology information.

Using a nested domain simulation that reduces from three nested simulations in the configuration (EC_LE ) to only one in the configuration (HA_LE ) to reach the concentration simulations at 0.025°as the objective can provide significant com-
putational benefits. By comparing the performance of the new approach with the traditional three-nesting method, we found that the computational cost was reduced by a factor of four while maintaining comparable accuracy in the results. This was achieved because the resolution of HARMONIE ensured that the boundary conditions were more comparable in terms of spatial resolution and was doable to go directly to the simulation objective grid. The reduction in the number of nested domains led to a substantial reduction in the computational resources required for the simulation, enabling us to tackle larger and more
complex problems with the same resources. Overall, the results of our study highlight the significant benefits of using a nested domain simulation with fewer levels of nesting and demonstrate its potential as a powerful tool for numerical simulations. HARMONIE operational data files are provided in 'grib' format. Standard and freely available. Each hourly gribfile has a file size of  200 Mb. Over two days, 16 runs are performed for each hour. Only 1/16th of the data volume provided will be needed to drive a CTM (  5 Gb / day) for a given forecast lead time and time window.

## 2.3   Cabauw meteorology information

The 213-meter tall KNMI-mast Cabauw generates continuum and stable meteorological measurements for a location with homogeneous characteristics in a central part of the Netherlands. This site is located in a flat terrain with an elevation of 0 meters above sea level and has been used to validate models, satellite information, and other meteorological sensors (Bosveld et al., 2020). The surrounding area is mainly used for agriculture purposes; although the Cabauw tower is located in a rural area, small
towns and villages are nearby. The data for this experiment was downloaded from https://dataplatform.knmi.nl/dataset/cesar-tower-meteo-lb1-t10-v1-2 for the months April-May-June-July-August 2019. The data comes in 10 minutes Interval of sampling and contains the following variables: Air temperature, Dew point temperature, Specific humidity, Wind speed, and wind direction.

### 2.3.1   Surface concentration pollutants information

The $NO_2$ data was downloaded from the ground base sensor stations of different from (www.luchtmeetnet.nl). The different locations along the country were chosen to compare the two $NO_2$ in the LOTOS-EUROS model configuration to cover the more representation possible





## 2.4 TROPOMI

The TROPOMI information was explored qualitatively because we wanted to establish a period from which we can have some
well-defined characteristics to have a priory knowledge of the concentration state at the tropospheric and total column level, at
least for the daily satellite snapshot.

## 3 Results

### 3.1 Meteorology fields evaluation

Figure (2) compares the temperature ECMWF meteorology and the temperature Cabauw observations at different levels, as
well as an image from the Cabauw tower illustrating the sensor positions for comparison with the ECMWF and HARMONIE
models. Panel (a) of the figure displays the time series of temperature from the ECMWF meteorology compared with the
temperature Cabauw observations at different levels. The comparison shows some differences between the two datasets at
certain levels, particularly during nighttime; the daily cycle is in phase, but there are differences in magnitudes. This suggests
the importance of validating model outputs with ground-based observations.

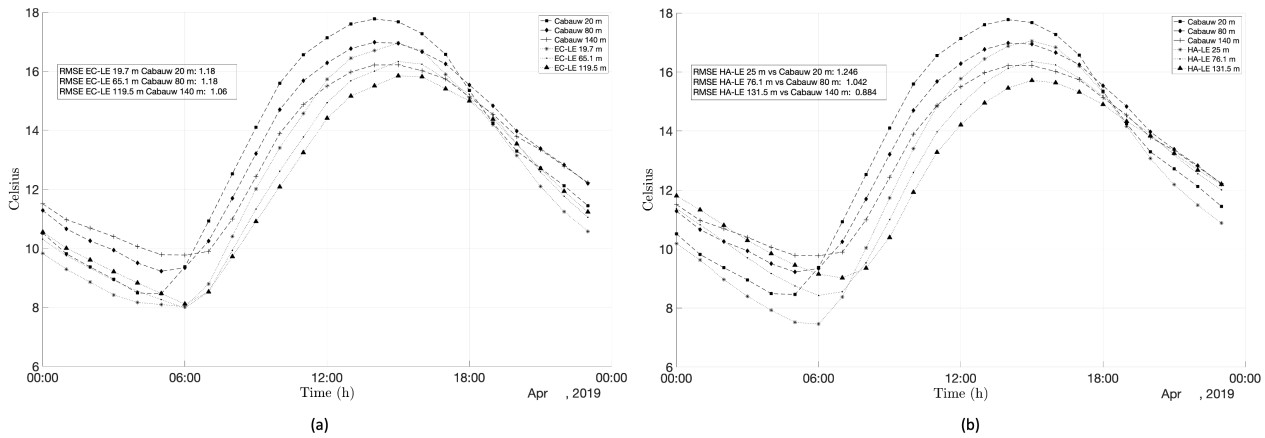

**Figure 3.** The daily temperature cycle from ECMWF (a) and HARMONIE (b) models and Cabauw observations at different LOTOS-EUROS
simulation levels. The RMSE for different levels is shown for the two input meteorological value compared against the sensors in the tower

Panel (b) of the figure provides an image from the Cabauw tower, with the positions of the sensors and the ECMWF and
HARMONIE models overlaid in three different colors to illustrate the height of the levels for comparison. This information is
essential for validating the models' height levels and identifying potential sources of discrepancies between the model outputs
and the observations in the height structure.

Overall, the results in Figure (2) demonstrate the importance of validating model outputs with ground-based observations and
the value of visualizing sensor positions and model outputs together for comparison. These findings can inform improvements



to the models and ultimately lead to more accurate temperature and other meteorological variables predictions. Figure (3) shows the daily cycle for three levels of the two meteorology input information to the LOTOS-EUROS model compared with the respective height sensor in the Cabauw tower. The comparable values show minor differences, which gives technical trust in the model configuration. For the height of 140 m from the Cabauw tower, the HARMONIE meteorology got a lower RMSE, showing better agreement with the measurements in the extreme part of the day.

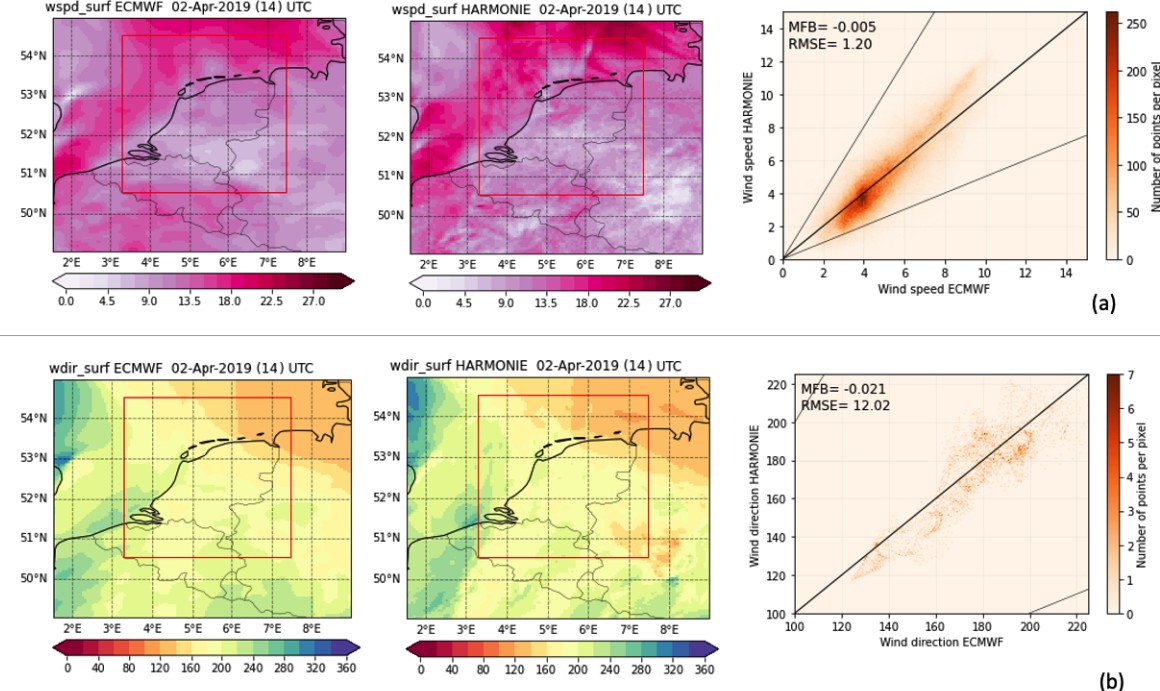

**Figure 4.** Instantaneous spatial comparison between the surface wind speed [m/s] (wspd_surf) and direction [°] (wdir_surf) interpolated to the simulation resolution grid, and in the right image, a quantitative comparison in the red square demarcated over The Netherlands where the RMSE and the MFB scores are shown. Base maps from http://www.gadm.org/

In Figure (4) can we see a spatial comparison of the wind direction and magnitudes at the resolution of the model simulation, and on the right side of the image, some more statistical comparisons based on different metrics of this variable comparison over the red square over there map. When comparing the results, we found that the overall performance was comparable. However, there were some differences in the details of these fields. These differences may have contributed to variations in the results observed between studies.

Despite the performance similarities, further investigation is needed to determine the most effective approach for achieving accurate results. Other results of validation of the meteorological variables of the HARMONIE model but in this case from the Dutch Offshore Wind Atlas (DOWA) against Cabauw vertical measurements, can be found in (Knoop et al., 2020)



### 3.2 Concentration fields validation

We compared the surface concentration of $NO_2$ for the (EC_LE ) and (HA_LE ) configurations and visualized the results in Figure (5). Panel (a) of the figure shows the surface concentration of $NO_2$ for the (EC_LE ) configuration, while panel (c) shows the surface concentration of $NO_2$ for the (HA_LE ) configuration.

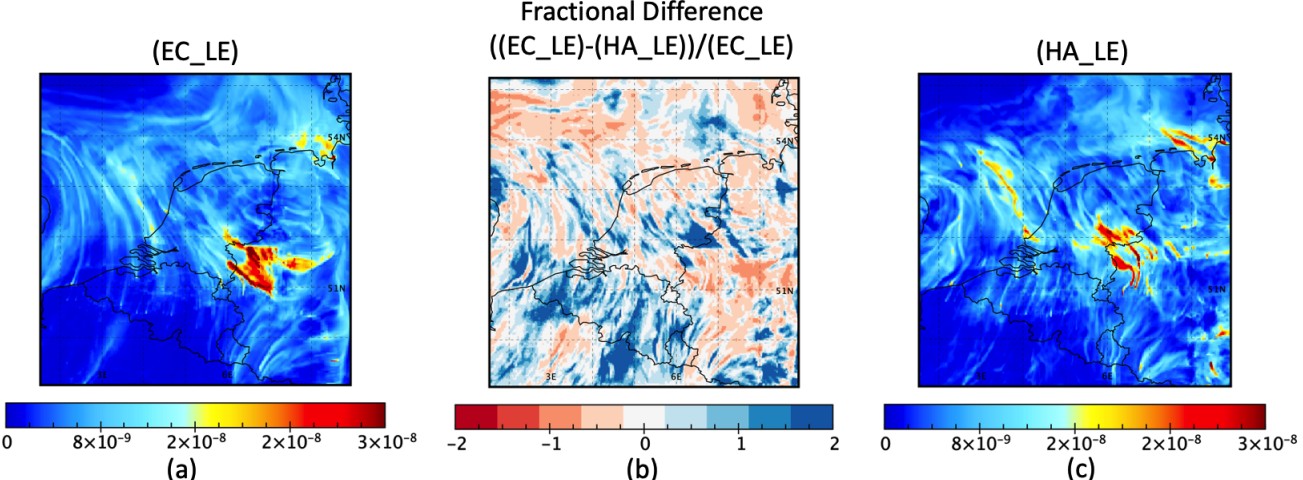

**Figure 5.** Air masses distinctions from the comparisons for the system configurations in volume mixing ratio of surface $NO_2$ [mol mol$^{-1}$] from (a) (EC_LE ) and (c) (HA_LE ). The middle panel (b) shows the fractional difference. Base maps from http://www.gadm.org/

To gain further insights into the differences between the two configurations, we included a difference comparison in panel (b). The difference comparison ((EC_LE )-(HA_LE ))/(HA_LE ) clearly shows that the (HA_LE ) configuration produces
different $NO_2$ concentrations than the (EC_LE ) configuration at the air mass of specific locations, revealing a wind direction difference indicated by the bias observed in the plumes depending on the meteorology uses to drive each model which can impact the time series in any location. This finding suggests that wind direction can play a crucial role in the transport and diffusion of $NO_2$ in the atmosphere and can affect the accuracy of the modeled concentrations. This experiment shows air mass characterization based on $NO_2$ concentration plume structures. The statistical metric lets us quantify the areas where the
HA_LE overestimates the EC_LE , indicating the discrepancy between the two sources of information.

The fractional difference Specifically, the direction of the wind can influence the transport of $NO_2$ emissions from their sources to other areas, leading to variations in the concentrations of the pollutant.

Our results provide insights into the factors contributing to variations in $NO_2$ concentrations in the Netherlands and underscore the need to carefully consider model configurations with meteorological input in atmospheric chemistry modeling. The
tropospheric column of $NO_2$ for the (EC_LE ) and (HA_LE ) configurations, as well as the TROPOMI satellite retrieved information for this pollutant for the troposphere, are shown in Figure (6). Panel (a) of the figure shows the tropospheric column of



NO$_2$ for the (EC_LE ) configuration, while panel (b) shows the tropospheric column of NO$_2$ for the (HA_LE ) configuration. Panel (c) shows the tropospheric column of NO$_2$ obtained from the TROPOMI satellite retrieval.

The comparison reveals that the (HA_LE ) configuration produces a tropospheric column of NO$_2$ that is more similar to
the TROPOMI satellite retrieval, particularly in regions with high NO$_2$ concentrations. This similarity is likely due to a slight change in wind direction in the HARMONIE configuration, which affects the transport and diffusion of NO$_2$ emissions in the atmosphere. In addition to revealing differences in NO$_2$ concentrations between the two configurations and the satellite retrieval, the images in Figure (6) show different details over the maps. Specifically, the maps illustrate the locations of coal and gas power energy stations, oil rigs and pipelines, principal airports, and roads across the Netherlands. These details are
important to consider in atmospheric chemistry modeling, as they can help to identify potential sources of NO$_2$ emissions and inform policy decisions related to air quality management.

Figure 7 presents a comprehensive analysis of air quality measurements obtained from three stations within the lucht-meetnet.nl network. The stations, namely Utrecht Kardinaal de Jongweg (a), Rotterdam Zuid-Pleinweg (b), and Valthermond Noorderlep (c), are compared against two model configurations depicted in the upper panel. The first configuration, ECMWF-
>LOTOS-EUROS, is visualized in orange, while the second configuration, HARMONIE->LOTOS-EUROS, is depicted in blue. The evaluation focuses on the representative error in dispersion, specifically examining the deviation of the grid cell where each station is located and its immediate neighboring cells. The lower panel of the figure compares the surface Kz coefficient, offering additional insights into the analysis of air quality data.

In panel (a), the (EC_LE ) configuration shows lower NO$_2$ concentrations in some areas compared to panel (b), where
the (HA_LE ) configuration produces higher NO$_2$ concentrations in the same regions. These differences may be attributed to using different meteorological and emission data in the two configurations, which can affect the model's ability to simulate atmospheric chemistry accurately.

Overall, comparing the two configurations highlights the importance of carefully selecting appropriate model configurations when evaluating NO$_2$ concentrations in a given region. Further research is needed to investigate the specific factors contributing
to the differences between the two configurations and determine which configuration is more accurate for NO$_2$ concentration modeling in the Netherlands. The transversal cut over the Netherlands in Figure (8) shows a comparison between the (EC_LE ) configuration in the upper panel and the (HA_LE ) NO$_2$ fields in the panel below. The figure indicates notable differences in the NO$_2$ concentration fields produced by the two configurations in the columns and the value of the $K_z$ diffusion coefficient at the layer interfaces. The planet boundary layer is shown in all pictures with a shaded blue line. Here, the HARMONIE provides
a more complex structure that must prevail in the impact of vertical modeled transport.

Figure 9 compares both configurations for a mean of April for 4 levels of the NO$_2$ concentration and the diffusion coefficient.

The HARMONIE atmospheric model stands out with its enhanced structure and distinct field shape compared to the ECMWF. However, it exhibits a discrepancy when simulating the boundary layer height, overestimating it compared to real-world observations. This disparity significantly affects air pollutant concentrations, particularly in the upper atmosphere. The
higher simulated boundary layer height in HARMONIE allows pollutants to be transported to higher altitudes, leading to complex chemical reactions and the formation of secondary pollutants. This phenomenon affects regional air quality, climate, and





**Figure 6.** Comparison between the tropospheric column of NO$_2$ (EC_LE) (a) and(HA_LE) (b) for the TROPOMI tropospheric column at the overpass time (c). Different characteristics are shown in the figures such as the power plants, principal airports and roads. The ground measurement station depicted with a star are the stations shown in the next figure. Units are different in the model and satellite column concentration shown but for the purpose of the comparisons the plume structure and direction is the intended. Base maps from (http://www.gadm.org/) and information from (https://emodnet.ec.europa.eu/en/human-activities)

the understanding of long-range pollutant transport. Accurately representing the boundary layer height is crucial for reliable air quality forecasts and assessing pollutant impacts. Resolving this issue requires further research and refinement of the model's parameterizations and processes related to boundary layer dynamics, enabling improved simulations of pollutant dispersion in different atmospheric layers.




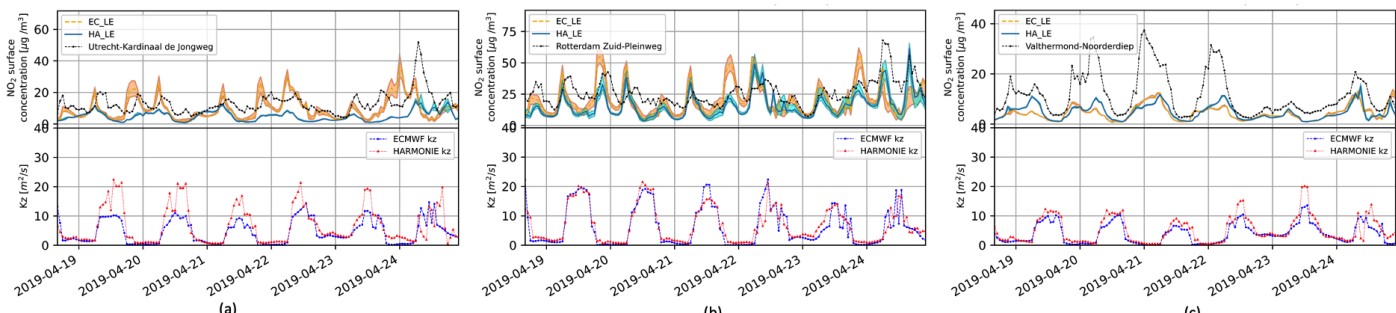

**Figure 7.** Three air quality stations from the (www.luchtmeetnet.nl) network (a) Utrecht Kardinaal de Jongweg (b) Rotterdam Zuid=-Pleinweg (c) Valthermond Noorderlep compared with the two model configurations in the upper panel (ECMWF->LOTOS-EUROS in orange and HARMONIE->LOTOS-EUROS in blue) taking the representative error in the dispersion such as the deviation of the grid cell where the station is located and the immediate cells around. The below panel shows a comparison for the surface Kz coefficient

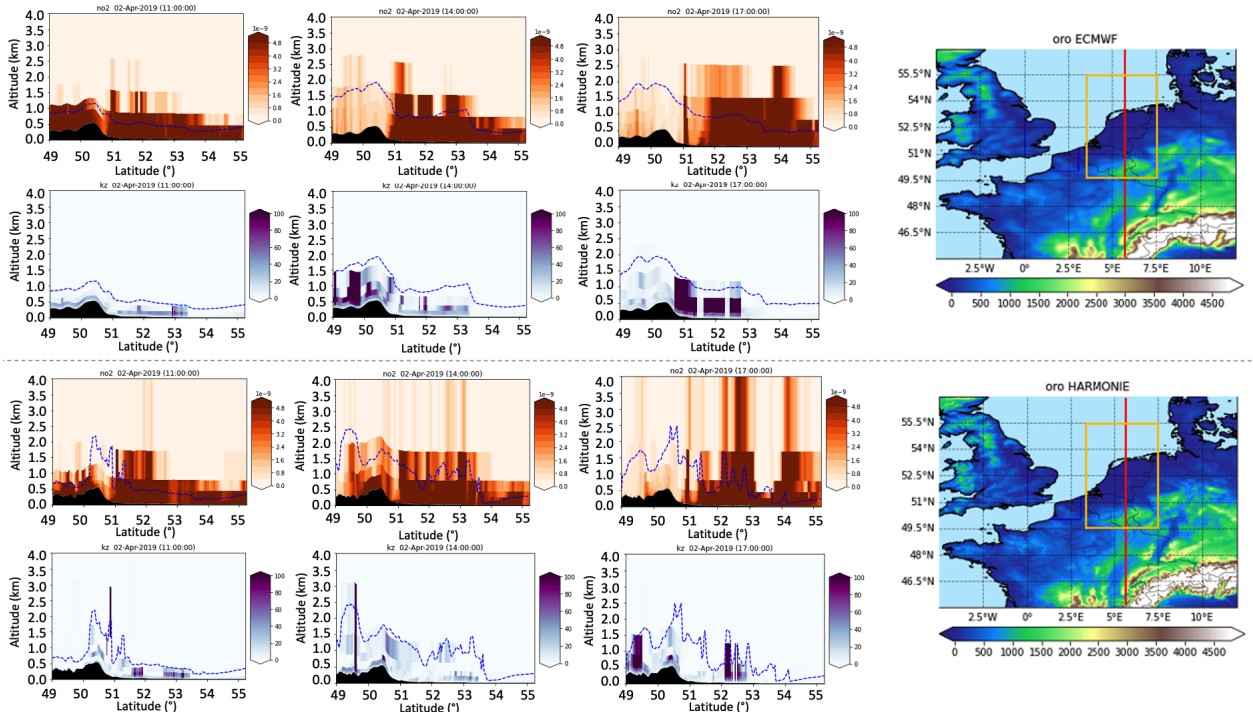

**Figure 8.** (a) Transversal cuts on longitude (6.2 ° E ) over the Netherlands comparison between the (EC_LE ) configuration, and (b) the (HA_LE ) NO$_2$ concentration fields. The dashed blue lines correspond to the planetary boundary layer in the models. The panels on the right show each of the transversal cuts. Base maps from (http://www.gadm.org/)





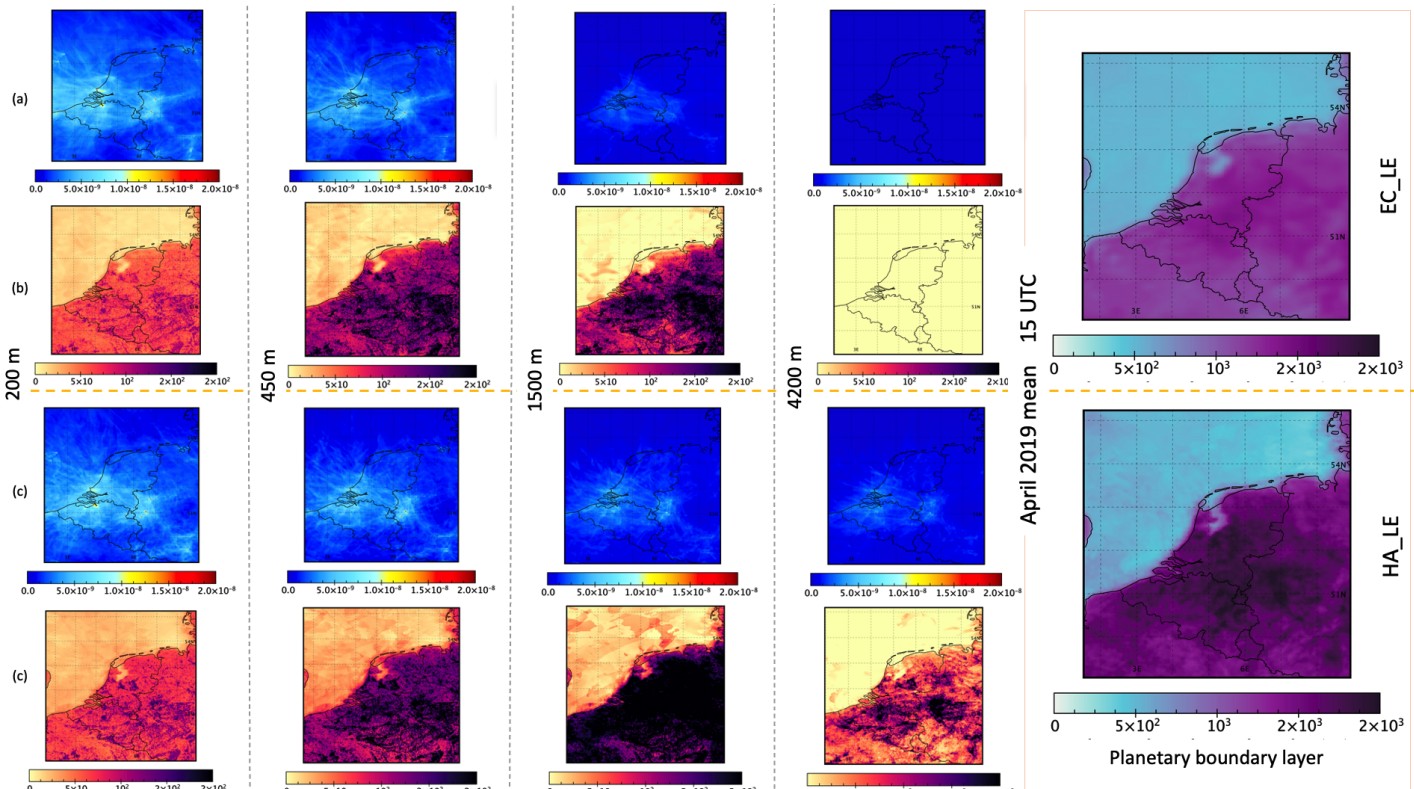

**Figure 9.** April mean (15 UTC) NO$_2$ concentration fields [mol mol$^{-1}$] and Kz [m$^{[2]}$ s$^{-1}$] at 200, 450, 1500, and 4200 m altitude (a,c) for EC_LE and (b,d) for HA_LE. Base maps from (http://www.gadm.org/)

## 4 Discussion

The hydrostatic nature of a meteorological model refers to the assumption that the atmosphere is in a state of hydrostatic equilibrium, meaning that the vertical pressure gradient balances the gravitational force. In this configuration, the atmospheric equations used by the model do not include the effects of non-hydrostatic processes, such as wind, turbulence, and gravity

waves. In contrast, a non-hydrostatic meteorological model allows for including non-hydrostatic processes in the atmospheric equations. This can provide a more accurate representation of the dynamics of the atmosphere, especially in regions where these processes are significant, such as near the coast, over forests, and in urban areas.

The choice of a hydrostatic or non-hydrostatic meteorological configuration can significantly impact the performance of a chemical transport model. A hydrostatic configuration may be sufficient in some cases, but a non-hydrostatic configuration

may be necessary to represent the transport of pollutants in the atmosphere accurately. Overall, it is essential to carefully consider the meteorological model's capabilities and the study region's specific characteristics when choosing a hydrostatic




or non-hydrostatic configuration for a chemical transport model. This can ensure that the model can accurately represent the transport and impact of pollutants on air quality.

The vertical velocity fields in the LOTOS-EUROS model are calculated using the convergence and divergence of the horizontal winds from the meteorological model. This allows the model to simulate the effects of vertical motion in the atmosphere on pollutants' transport and chemical reactions. The availability of vertical meteorological fields can impact the accuracy and reliability of the LOTOS-EUROS model's predictions. If vertical wind data is unavailable or is of low quality, the model may not accurately represent the vertical motion of pollutants in the atmosphere. This can lead to errors in the model's predictions of the distribution and impact of pollutants on air quality. Other models, such as CHIMERE, recently evaluated the vertical mechanism to improve also the vertical transport (Menut et al., 2021). To improve the performance of the LOTOS-EUROS model, it is crucial to ensure that high-quality vertical wind data is available from the meteorological model. This can provide more accurate and realistic representations of the vertical motion of pollutants in the atmosphere and improve the accuracy of the model's predictions.

Using high-resolution meteorology in a chemical transport model like LOTOS EUROS can improve the accuracy and reliability of the model's predictions. High-resolution meteorological data provides more detailed information about the atmosphere's wind, temperature, pressure, and humidity conditions, which can be used to simulate the movement of pollutants and trace gases more accurately. In particular, high-resolution meteorology can provide more accurate representations of the effects of small-scale atmospheric processes, such as turbulence and convection, on pollutant transport and chemical reactions. This can improve the model's ability to simulate the distribution and impact of pollutants on air quality and can provide more detailed and helpful information for air quality forecasting and environmental management.

The following step is the preparation for assimilation from the side of the data and the model perspective. Figure (10) shows the two products needed to assimilate. These results highlight the importance of carefully considering model configurations and meteorological factors in atmospheric chemistry modeling and the potential benefits of satellite remote sensing data in improving the accuracy of the modeled $NO_2$ concentrations. The comparison between the LOTOS-EUROS simulated retrieval of the tropospheric column of $NO_2$ Ys and the TROPOMI average tropospheric vertical column Yr product from the CSO preprocessing tool that is the input needed for the data assimilation stage is shown in Figure (10).

Panel (a) of 10 shows the LOTOS-EUROS simulated retrieval of the tropospheric column of $NO_2$ Ys, while panel (b) shows the TROPOMI average tropospheric vertical column Yr product. The comparison indicates that there are significant differences between the two products, particularly in regions where there are high $NO_2$ concentrations. These differences are important to consider in the data assimilation stage, as they can impact the accuracy of the assimilated data and, ultimately, the accuracy of the analysis modeled $NO_2$ concentrations.

Using high-resolution meteorology in chemical transport models like LOTOS EUROS can provide valuable insights into the transport and impact of pollutants on air quality and support decision-making and policy development to improve air quality and protect public health.

For data assimilation, it is essential to get estimations of the accuracy of the observations to construct the observation error covariance matrix; the error from the observations is used to build a diagonal matrix $R$ because the error values at this stage are



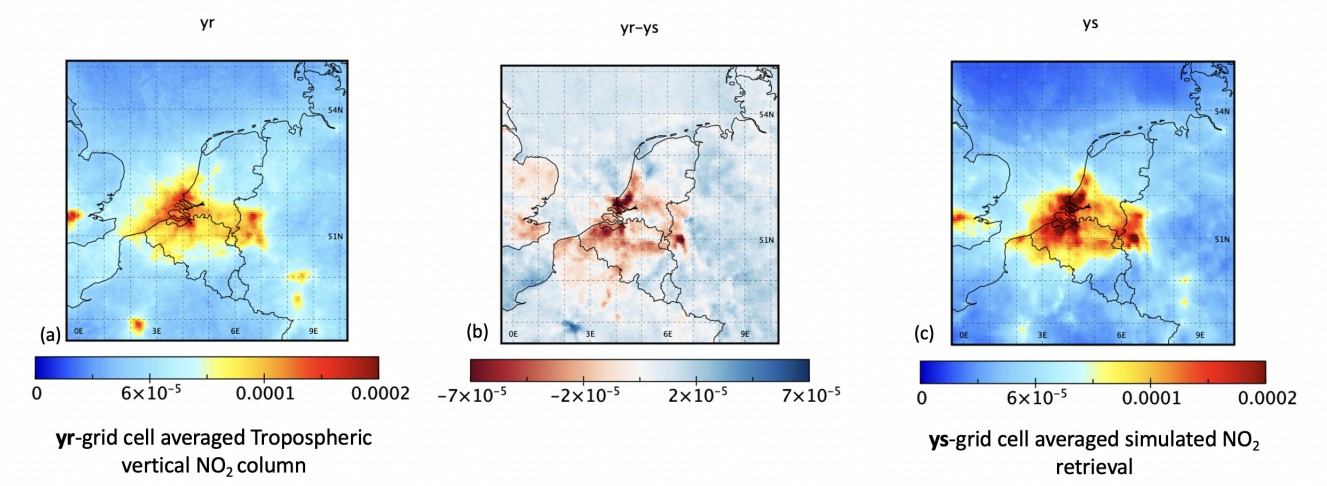

**Figure 10.** Comparison between the LOTOS-EUROS simulated retrieval of the tropospheric column of $NO_2$ Ys and the TROPOMI average tropospheric vertical column Yr product from the CSO preprocessing tool. Base maps from (http://www.gadm.org/)

correlated only with the observed state in the already remapped grid. The inaccuracies in the TROPOMI observations result from the retrieval method's three stages, which are a previous step in pre-processing the satellite information from manipulating the crude light spectroscopy data to have the $NO_2$ vertical column density. The stages that add errors in this process are the
quantification of slant columns, the separation of the stratospheric and tropospheric components of slant columns, and the tropospheric air mass factors multiplication (Van Geffen et al., 2020). The overall error is provided per pixel in the TROPOMI data product.

## 5   Conclusions

The HARMONIE (cy43) coupling with LOTOS-EUROS mimicking the ECMWF with LOTOS-EUROS technically works,
showing comparable results in meteorology variables and $NO_2$ concentrations. Differences in the details can be perceived mostly in the vertical column concentration, for which in the HARMONIE configuration, highly values appear in the upper layer of the atmosphere than in the ECMWF configuration, which was caused for the differences in the vertical diffusion co-efficient. The HARMONIE atmospheric model stands out with its enhanced structure and distinct field shape compared to the ECMWF. However, it exhibits a discrepancy when simulating the boundary layer height, overestimating it compared to real-
world observations. This disparity significantly affects air pollutant concentrations, particularly in the upper atmosphere. The higher simulated boundary layer height in HARMONIE allows pollutants to be transported to higher altitudes, leading to complex chemical reactions and the formation of secondary pollutants. This phenomenon affects regional air quality, climate, and the understanding of long-range pollutant transport. Accurately representing the boundary layer height is crucial for reliable air quality forecasts and assessing pollutant impacts. Resolving this issue requires further research and refinement of the model's

parametrizations and processes related to boundary layer dynamics, enabling improved simulations of pollutant dispersion in
different atmospheric layers; so far, inconclusive concerning performance in the surface concentrations compared with ground
stations. The fields evaluated (meteorology and NO2 concentrations) are comparable, with no significant improvement in sur-
face NO2 compared to observations at surface stations. There is potential to further develop LOTOS-EUROS at high spatial
resolution in the HARMONIE configuration because of the less work in nesting domains to simulate at least the resolution
objective in this work properly (0.025 °). The next step in this work is to use both configurations, ECMWF and HARMONIE,
in the data assimilation experiment of TROPOMI NO2 using LOTOS-EUROS to understand the impact of this non-hydrostatic
meteorology in the transport of contaminants.

*Author contributions.* Conceptualization, Andrés Yarce Botero and Michiel van Weele; methodology, Andrés Yarce Botero; Andrés Yarce
Botero software; Andrés Yarce Botero and Arjo Segers validation; Andrés Yarce Botero and Michiel van Weele and Arjo Segers and Henk
Eskes formal analysis; Michiel van Weele and Henk Eskes and Pier Siebesma resources; Andrés Yarce Botero data curation; Andrés Yarce
Botero writing original draft preparation; Michiel van Weele writing review and editing; visualization; Michiel van Weele project adminis-
tration. All authors have read and agreed to the published version of the manuscript.

## Appendix A:  Appendix

*Competing interests.*  Authors declare that no competing interests are present

## 335   Code availability

The codes are available at the GitLab repository https://ci.tno.nl/gitlab/lotos-euros/le-harmonie/-/tree/Andres_branch

*Acknowledgements.*  Andres Yarce Botero is supported by the NWO program gebruikersondersteuning under grant KNW19002 (Dutch
collaborative network for air pollution monitoring using satellites)

. Map data copyrighted by OpenStreetMap contributors and available from https://www.openstreetmap.org



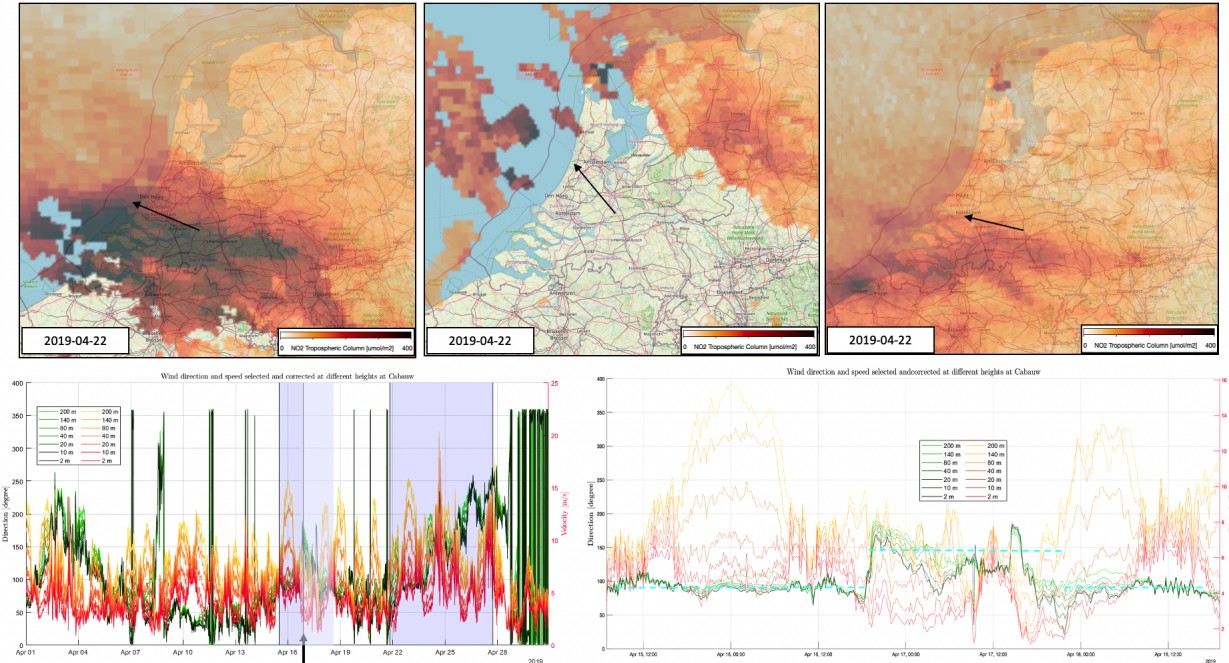

**Figure A1.** Transport plumes of NO$_2$ TROPOMI Tropospheric column observations compared with the CABAUW observations for wind direction and magnitude for 7 levels from 2 m to 200m. © OpenStreetMap contributors 2021. Distributed under the Open Data Commons Open Database License (ODbL) v1.0.

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



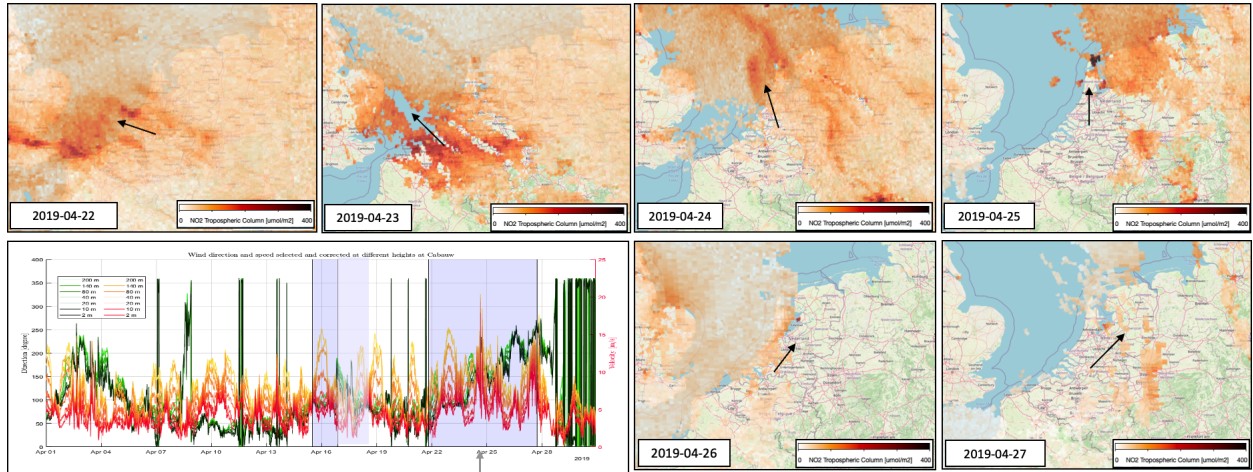

**Figure A2.** Transport plumes of NO$_2$ TROPOMI Tropospheric column observations compared with the CABAUW observations for wind direction and magnitude for 7 levels from 2 m to 200m from 2019-04-22 to 2019-04-27 in which a scenario of changing air mass direction drive the transport of contaminants. © OpenStreetMap contributors 2021. Distributed under the Open Data Commons Open Database License (ODbL) v1.0.

Clark, P., Roberts, N., Lean, H., Ballard, S. P., and Charlton-Perez, C.: Convection-permitting models: a step-change in rainfall forecasting, Meteorological Applications, 23, 165–181, 2016.

Ding, J.: Impact of HARMONIE high-resolution meteorological forecasts on the air quality simulations of LOTOS-EUROS, Royal Netherlands Meteorological Institute, 2013.

El-Harbawi, M.: Air quality modelling, simulation, and computational methods: a review, Environmental Reviews, 21, 149–179, 2013.

Engdahl, B. J. K., Thompson, G., and Bengtsson, L.: Improving the representation of supercooled liquid water in the HARMONIE-AROME weather forecast model, Tellus A: Dynamic Meteorology and Oceanography, 72, 1–18, 2020.

Escudero, M., Segers, A., Kranenburg, R., Querol, X., Alastuey, A., Borge, R., de la Paz, D., Gangoiti, G., and Schaap, M.: Analysis of summer O 3 in the Madrid air basin with the LOTOS-EUROS chemical transport model, Atmospheric Chemistry and Physics, 19, 14 211–14 232, 2019.

Fountoukis, C. and Nenes, A.: ISORROPIA II: a computationally efficient thermodynamic equilibrium model for K+– Ca2+– Mg2+– NH4+ – Na+– SO4 2− – NO3< su p>, Atmos. Chem. Phys, 7, 4639–4659, 2007.

Gibbon, J. and Holm, D. D.: Extreme events in solutions of hydrostatic and non-hydrostatic climate models, Philosophical Transactions of the Royal Society A: Mathematical, Physical and Engineering Sciences, 369, 1156–1179, 2011.

Haakenstad, H., Breivik, Ø., Furevik, B. R., Reistad, M., Bohlinger, P., and Aarnes, O. J.: NORA3: A nonhydrostatic high-resolution hindcast of the North Sea, the Norwegian Sea, and the Barents Sea, Journal of Applied Meteorology and Climatology, 60, 1443–1464, 2021.

Kalverla, P., Steeneveld, G.-J., Ronda, R., and Holtslag, A. A.: Evaluation of three mainstream numerical weather prediction models with observations from meteorological mast IJmuiden at the North Sea, Wind Energy, 22, 34–48, 2019.



Khan, S. and Hassan, Q.: Review of developments in air quality modelling and air quality dispersion models, Journal of Environmental Engineering and Science, 16, 1–10, 2020.

Knoop, S., Ramakrishnan, P., and Wijnant, I.: Dutch Offshore Wind Atlas Validation against Cabauw Meteomast Wind Measurements, Energies, 13, 6558, 2020.

Lorenc, A. C. and Jardak, M.: A comparison of hybrid variational data assimilation methods for global NWP, Quarterly Journal of the Royal Meteorological Society, 144, 2748–2760, 2018.

Manders, A. M., Builtjes, P. J., Curier, L., Denier van der Gon, H. A., Hendriks, C., Jonkers, S., Kranenburg, R., Kuenen, J. J., Segers, A. J., Timmermans, R., et al.: Curriculum vitae of the LOTOS–EUROS (v2. 0) chemistry transport model, Geoscientific Model Development, 10, 4145–4173, 2017.

Manders-Groot, A., Schaap, M., van Ulft, B., and van Meijgaard, E.: Coupling of the air quality model Lotus-Euros to the climate model Racmo, National Research Programme Knowledge for Climate Nationaal Onderzoekprogramma Kennis voor Klimaat (KvK) All rights reserved, 2011.

Marseille, G.-J. and Stoffelen, A.: Toward Scatterometer Winds Assimilation in the Mesoscale HARMONIE Model, IEEE Journal of Selected Topics in Applied Earth Observations and Remote Sensing, 10, 2383 – 2393, https://doi.org/10.1109/JSTARS.2016.2640339, cited by: 10, 2017.

Menut, L., Bessagnet, B., Briant, R., Cholakian, A., Couvidat, F., Mailler, S., Pennel, R., Siour, G., Tuccella, P., Turquety, S., et al.: The CHIMERE v2020r1 online chemistry-transport model, Geoscientific Model Development, 14, 6781–6811, 2021.

Pielke, R. A. and Uliasz, M.: Use of meteorological models as input to regional and mesoscale air quality models—limitations and strengths, Atmospheric environment, 32, 1455–1466, 1998.

SAITO, K., ichi ISHIDA, J., ARANAMI, K., HARA, T., SEGAWA, T., NARITA, M., and HONDA, Y.: Nonhydrostatic Atmospheric Models and Operational Development at JMA, Journal of the Meteorological Society of Japan. Ser. II, 85B, 271–304, https://doi.org/10.2151/jmsj.85B.271, 2007.

Schaap, M., Van Loon, M., Ten Brink, H., Dentener, F., and Builtjes, P.: Secondary inorganic aerosol simulations for Europe with special attention to nitrate, Atmospheric Chemistry and Physics, 4, 857–874, 2004.

Schaap, M., Timmermans, R. M., Roemer, M., Boersen, G., Builtjes, P., Sauter, F., Velders, G., and Beck, J.: The LOTOS? EUROS model: description, validation and latest developments, International Journal of Environment and Pollution, 32, 270–290, 2008.

Thürkow, M., Kirchner, I., Kranenburg, R., Timmermans, R., and Schaap, M.: A multi-meteorological comparison for episodes of PM10 concentrations in the Berlin agglomeration area in Germany with the LOTOS-EUROS CTM, Atmospheric Environment, 244, 117 946, 2021.

Van Geffen, J., Boersma, K. F., Eskes, H., Sneep, M., Ter Linden, M., Zara, M., and Veefkind, J. P.: S5P TROPOMI NO2 slant column retrieval: method, stability, uncertainties and comparisons with OMI., Atmospheric Measurement Techniques, 13, 2020.

van Stratum, B., Theeuwes, N., Barkmeijer, J., van Ulft, B., and Wijnant, I.: A One-Year-Long Evaluation of a Wind-Farm Parameterization in HARMONIE-AROME, Journal of Advances in Modeling Earth Systems, 14, e2021MS002 947, 2022.

Verzijlbergh, R.: Atmospheric flows in large wind farms, Europhysics News, 52, 20–23, 2021.

Viana Jiménez, S. and Díez Muyo, M. V.: Procesos de superficie en Harmonie-Arome y su importancia en procesos atmosféricos, Sexto simposio nacional de prediccion- Memorial Antionio Mestres, 2019.

Wichink Kruit, R., Schaap, M., Sauter, F., Van Zanten, M., and Van Pul, W.: Modeling the distribution of ammonia across Europe including bi-directional surface–atmosphere exchange, Biogeosciences, 9, 5261–5277, 2012.