# Peer review of "Investigating the impact of coupling HARMONIE-WINS50 (cy43) meteorologie to LOTOS-EUROS (v2.2.002) on simulation of $NO_2$ concentrations over The Netherlands"

_EGUsphere, 2023_

## Author Comment (AC4)

[revised manuscript text omitted]

Figure 7 presents a comprehensive analysis of air quality measurements obtained from three stations within the luchtmeetnet.nl network. The stations, namely Utrecht Kardinaal de Jongweg is located in a central part of the country (a), Rotterdam Zuid-Pleinweg located in a region more the west which correspond to the Rotterdam region known by the huge levels of pollutants due to the harbor and refineries activities (b), and Valthermond Noorderlep (c) which is located in a more rural area, 
[revised manuscript text omitted]

---

## Author Comment (AC6)

**Major**

1. What is the objective of this paper? Testing the meteorological forcing or the air quality simulation? This is not very clear and should be specified

We configure the coupling of new meteorology with high resolution (HARMONIE WINS50) to serve as an input information for the Chemical Transport Model (CTM) LOTOS-EUROS model. The meteorology fields generated from a Numerical Weather Model tend to come in a variety of data structures, shapes and variables that needs an effort having it ready to be used to drive a CTM. We describe the methodology we took for use this new available meteorology which has benefits compared to the used by default, as well points on the table new points to discuss to probably get the best impact possible such as the need of a vertical diffusion routine in the LOTOS-EUROS which uses explicitly the velocity in the vertical direction.

2. Section 2.1, what drives the model, ECMWF, or NWP fields from the other model? In the Introduction, you stated the latter, but here you describe ECMWF!

We describe both sources of meteorology because we wanted to present the effects of upgrading in resolution and in a meteorology of other nature (non -hydrostatic). The ECMWF was described as the default meteorology, which was used as the default standard to configure the data from the HARMONIE meteorology in the same way to have comparable scenarios.

3. Table 2: why there are two meteorological fields in LOTUS –EUROS?

Because the purpose of this paper is the comparison between LOTOS-EUROS output simulations with inputs from different meteorologies.

4. Why there are large differences between measurements and model results (Figure7)?

We saw differences between measurements and model results more comparable in the first levels than in the higher levels. The big differences here are due to the emission inventories used for this simulations which leads the underestimation of the models for example in Figure _7.

5. There are some language problems. Some are listed below, Please correct them.

**Minor**

| Reviewer comment | Action performed by the authors |
| --- | --- |
| L25-28: The same lines are in the abstract. Rephrase them | **Rephrase the lines in the paragraph with "Numerical Weather Prediction Models (NWP) supply the data required by Chemical Transport Models (CTM) to resolve the emission, transportation, chemical reactions and other atmospheric interactions of pollutants throughout the spatio-temporal field of interest"** |
| L30: Which CTM? | **In this part I talked about a CTM generally** |
| L36. Delete "representation" | **Word deleted** |
| L38: delete "the simulated" | **Word deleted** |
| L43: by van Stratum et al. (2022) | **Modified as suggested** |
| L46: CTM, it's already abbreviated | **Modified as suggested** |
| L53: space after the bracket | **Modified as suggested** |
| L53: what do you mean by frequent coupling; there must be a time step for this | **Modified with two-way coupling, which is the kind of coupling which the chemical fields also have effects on the meteorological fields. The RACMO-LOTOSEUROS system had this bidirectional coupling opposite to the other systems** |
| L56: by Ding (2013) | **Modified as suggested** |
| L61: ", respectively" | **Modified as suggested** |

| | |
|---|---|
| L100: similar to | **Modified as suggested** |
| L104: for this study comes from the | **Modified as suggested** |
| L109: observations or measurements | **All the document was homogenized to observtations** |
| L111: What is SNELLIUS? | **The information of what is SNELLIUS, , The Dutch National Supercomputer accessible at (snellius.surf.nl)** |
| L124: emulate? | **We consider the term emulate for this coupling because we took the ECMWF-LOTOSEUROS as the default system for which we wanted to mimic the fields needed in the same structure, variable names and other characteristics. The HARMONIE fields were treated in the same way, configuring all the needed variables from this meteorology in the same way** |
| Figure 2: The sensor and model levels are different. But can't you interpolate the model results to the sensor levels? | **We compared in the paper with the nearest level for each vertical measurement because we were interested to see the performance against the same model level in the two systems** |
| L211: ", specifically" | **Modified as suggested** |
| L234-237: agree, but which model results are close to measurements or reasonable? | **Both have good performance in the surface layers in different regions of the country** |
| L244-245: "that must prevail in the impact of …" I do not understand this. Are you talking about the uncertainty of the model results? A bias in the model simulations? | **Changed this paragraph for:"Overall, comparing the two system configurations highlights the importance of carefully selecting appropriate model configurations when evaluating NO$_{2}$ concentrations in a given region with a given simulation** |

| | |
|---|---|
| | **resolution. More research is needed to investigate the specific factors that contribute to the differences between the two configurations and determine which configuration is more accurate for modeling NO$_{2}$ concentration in the Netherlands.** |
| Figure 6: Something is written on the maps, but is too small to read | **The name labels on the map where changed for numbers and an extra table was incorporated with names and the number labels to improve readability** |
| L274-275: So what were the concluding results from the CHIMERE comparison? | **The results are qualitatively comparable in the sense of reduction of excess vertical diffusion. We did not compare with this model but pinpoint that manage explicitly the vertical diffusion is a good step to consider in a Chemical Transport Model. LOTOS-EUROS for this use an scheme wich use the horizontal fields from the meteorology to implicitly use its to calculate the vertical wind directions.** |
| L276: so the ECMWF wind data are not good? | **We were not conclusive to say this. Both wind fields have different nature. One is hydrostatic the other not and this has an impact of how to treat lower scales phenomena.** |

---

## Author Comment (AC7)

General comments

The manuscript "Investigating the impact of HARMONIE-WINS50 (cy43) and LOTOS-EUROS (v2.2.002) coupling on $NO_2$ concentrations in The Netherlands" by Andres Yarce Botero et al. introduces the chemical transport model LOTOS-EUROS driven by reanalysis meteorological data from HARMONIE and ECMWF at different spatial resolutions. The authors evaluated the impacts of these two meteorological datasets on simulated $NO_2$ concentrations over the North Sea during April 2019. The authors also used some surface observations and satellite retrievals to validate the model performance.

In general, this manuscript fits the scope of the Geoscientific Model Development. However, it does have several drawbacks. The description of the observation datasets used in this study is too brief and lacks sufficient information. The authors should provide more details, such as specifying the TROPOMI dataset and including citations in Section 2. In the results section, the discussion and analysis are ambiguous. The authors included little statistics to validate the model performance and compare the simulated results between the two experiments. It's not very convincing to only analyse the results at a few time snapshots (Figures 4, 6, and 8). The authors should include more details on the discrepancies between the observed and simulated $NO_2$ concentrations, as well as the factors that contribute to the differences between the two experiments. The uncertainties in air quality simulations driven by the two meteorological datasets have not been quantified. The conclusion of this paper is unclear. It's not evident whether the use of more precise meteorological data could lead to a more accurate air quality simulation. Additionally, some figures should be re-organized to provide more effective information (Figures 2 and 6). The authors might uniform and enlarge the labels and legends in figures for better readability. Considering these issues, the reviewer recommends publication after major revisions. Please refer to the specific comments and technical corrections listed below.

Answer to the general comments:

In response to the reviewers' general comments, we have made significant revisions to our manuscript. To augment the descriptive information of the observation datasets used in our study, we elaborated on the methodology section, furnishing detailed insights into our evaluation process.

Section 2.3,1 included now the following description for the ground measurements:

"*The $NO_2$ data was downloaded from the ground stations of different places in the Netherlands from www.luchtmeetnet.nl. Different locations in the country were chosen to compare the two $NO_2$ LOTOS-EUROS systems with the different meteorologies in the most representative area possible. This data is provided by Rijksinstituut voor Volksgezondheid en Milieu (RIVM). The RIVM is accredited for air quality measurements of the substances $SO_2$, NO, $NO_2$, $O_3$, PM2.5, and PM10 by the Dutch "Raad voor Accreditatie (RvA)" according to NEN-EN-ISO/IEC 17025:2018*"

And section 2.4 the following text has been included for the TROPOMI data:

"*The TROPOspheric Monitoring Instrument (TROPOMI) is the satellite instrument on board the Copernicus Sentinel-5 Precursor (S5p) satellite. S5P is a low-Earth polar orbit satellite. The polar orbit and wide coverage of the scanner provide almost daily global coverage. The TROPOMI spatial pixel resolution is 5.5 x 3.5 $km^2$ and the $NO_2$ retrieval uses a wavelength range of 405–465 nm (spectral band 4). The TROPOMI instrument is a spectrometer sensing ultraviolet (UV), visible (VIS), near (NIR), and short-wavelength infrared (SWIR) wavelengths to monitor Ozone ($O_3$), Methane ($CH_4$), Formaldehyde (HCHO), Aerosol, Carbon Monoxide (CO), Nitrogen Dioxide ($NO_2$), and Sulphur Dioxide ($SO_2$). The Royal Netherlands Meteorological Institute (KNMI) created the TROPOMI $NO_2$ retrieval method based on the DOMINO $NO_2$ retrieval algorithm employed on the Ozone Monitoring Instrument (OMI) precursor instrument (Boersma et. al 2021). In this work, the information from this instrument was explored qualitatively to establish a period from which we can have some well-defined characteristics to have prior knowledge of the concentration state at the tropospheric and total column level, for the daily satellite snapshot.*"

Secondly, with regards to assessing uncertainties in air quality simulations driven by the two meteorological datasets, we have incorporated the suggested statistics in the figures indicated bythe reviewer, and created the supplement material with 9 new graphics to offer a more comprehensive validation of our model's performance:

I. Panel illustrating the comparison of the diurnal cycle of specific humidity, as measured at the Cabauw tower, with model levels obtained from both the ECMWF (a) and HARMONIE (b) models. The differences are quantified using the RMSE statistic.

II. *A comparison of the boundary layer and wind velocity of ECMWF and HARMONIE at various resolutions interpolated to a LOTOS-EUROS grid resolution indicating enhanced structural features in the HARMONIE data. The dissimilarities between ECMWF and HARMONIE are particularly strong in the North Sea regions where contrasting boundary layers are apparent.*

III. *Instantaneous comparison (b) of the boundary layer between ECMWF (a) and HARMONIE (c) indicating enhanced HARMONIE structural features for the 13:00 UTC*

IV. *Boundary layer height for the two meteorology inputs ECMWF (a) and HARMONIE (b) with a scatter plot (c) to quantify some statistics over the red square on the Netherlands regions*

V. *Surface wind direction and wind speed comparison for the two meteorology inputs and scatter plot to quantify some statistics over the red square on the Netherlands regions*

VI. *$NO_2$ concentration from the ground observation Dutch air quality network time series compared with the two model surface concentrations and the Boundary layer height for the grid cell where the surface stations are located*

VII. *$NO_2$ surface concentration is compared with the two surface concentrations in the model, along with the $K_z$ diffusion coefficient for the grid cell that displays the surface concentration.*

VIII. *The $NO_2$ surface concentration, as measured by the air quality network, is compared with the two surface concentrations in the model. In this case, we take into account the representative error for the simulated fields by considering the mean grids around the measurement and the corresponding standard deviation, as depicted in the time series of the model. The $K_z$ diffusion coefficient for the grid cell that displays the surface concentration is also included in the panel below.*

IX. *Wind direction and velocity data for the Cabauw tower sensors are presented for the period between 22nd and 28th April 2019. This image serves as a complement to Figure A2 in the appendix section of the manuscript, allowing us to identify a time window within this month where the air mass gradually shifted from west to east in a clockwise orientation.*

Thirdly, this paper aims to evaluate the differences in the meteorological driver of LOTOS-EUROS. In the figures we have added to the supplementary material we provide statistics to quantify these differences. For the time series comparing the $NO_2$ concentrations of

the model and the observations (Figure_7) we added for the quantification of the differences a in terms of three common statistics an extra legend in the upper subplot

Fourthly, the discrepancies in the vertical distribution of trace gases are mainly due to the specific coupling method used for HARMONIE or ECMWF meteorology with LOTOS-EUROS. The issue concerning vertical transport requires further examination and is considered an area for future research. Our revised conclusions highlight the necessity for additional validation with $NO_2$ profile measurements unavailable for this publication.

In addition, we have enhanced the clarity and accessibility of our figures by improving the labels and legends and introducing tables that present the information depicted in the figures more straightforwardly. We rectified the inconsistencies between Table 1 and its caption as highlighted by Reviewer 2, by amending these in the revised manuscript.

Specific comments:

| Reviewer comment | Reaction from the authors |
|---|---|
| P2, Line 43: Please add the citation for "HARMONIE cycle 43" or give more description on it. | New references were added to this paragraph where the HARMONIE information is depicted. (Bengtsson_2017, Van Stratum 2022, plus some references inside this papers) |
| P6, Table 1: "The variables are divided into static (purple), dynamic two (red), and three dimensions (green)". Why do you include colors to represent the different variables? | Originally, the table was made using colors; however, a black-and-white version was created in response to the first editing. The table caption still contained references to the colors; this was an oversight and has been corrected in the manuscript. |

| | |
|---|---|
| "The variables underlined were calculated with other available variables". I didn't see any variables underlined. Please make sure the caption of Table 1. | This was also an oversight that has been corrected in the manuscript by changing the wording within the HARMONIE column: "Calculated from…". |
| P5, Line 139: Why did the authors use three nested domains in EC_LE experiment? Why didn't the authors directly interpolate the original ECMWF meteorology to 0.025 deg and use one domain? I'm wondering how the LOTOS-EUROS runs three nested-domain simulation. | A factor of three might be a conservative approach to nested zooming, the most efficient strategy has not been investigated for this work and this standard practice of a factor three was used for LOTOS-EUROS in a similar way as in e.g (Escudero,2019).

LOTOS-EUROS sequentially runs the nested-domain simulation from the coarse to the fine resolution, withhe latter taking the initial concentrations from the coarse resolution simulation. |
| P8-9, Sections 2.3.1 and 2.4: The descriptions on surface $NO_2$ data and TROPOMI data are incomplete. Please give more information including the surface sites you selected, quality control on the raw data, the TROPOMI dataset you used in this study and the citations. | The $NO_2$ station's names and coordinates were added with an extra table in the section denoted.

The quality control of this data comes from the accreditation. The data supplier (RIVM) is accredited for air quality measurements of the substances SO2, NO, NO2, O3, PM2.5 and PM10 by the Dutch "Raad voor Accreditatie (RvA)" according to NEN-EN-ISO/IEC 17025:2018 |
| P7, Figure 2: I would suggest the authors to show the comparison between the observations and ECMWF meteorology data at similar altitudes (e.g. 20 m, 40 m, and 200 m) in Figure 2a. It's very hard to distinguish the observations and reanalysis data in current figure. Why didn't the authors add the HARMONIE meteorology data in Figure 2a? | We made this comparison at the nearest altitude because we wanted to make this comparison before the regridding performed by the Chemical Transport Model. Once the CTM takes this meteorology it interpolates the variables to the simulation levels ofthe model. We removed Figure 2a from the manuscript because the goal of this part of the manuscript (Methodology) is to illustrate characteristics of the experiments and not |

| | |
|---|---|
| | to compare results already; the later is done in for example the temperature comparisons in Figure 3. |
| It would be better if the authors can add the altitudes of the sensors, the ECMWF and HARMONIE levels in Figure 2b. | The altitude value of the sensors and the two meteorological models was added to Figure 2. |
| P9-10, Section 3.1: I would suggest the authors to include some statistics such as NMB, RMSE, and correlation coefficients between the observations and ECMWF/HARMONIE meteorology data in the main text. The current analysis is too ambiguous. | For Figure 7 in this section, the suggested statistics by the reviewer were added. Additionally, a new document wity supplementary material was generated with extra statistical information and more graphics comparing the observations and ECMWF or HARMONIE meteorology. |
| P9, Figure 3: Is this the monthly mean daily temperature cycle in April 2019? The symbols and texts in Figure 3 are too small to identify. | Yes, it is the monthly mean daily temperature cycle with the RMSE. The text size and marker size has been increased. |
| P11, Line 204: The relative difference in Figure 5b is defined as "((EC_LE )-(HA_LE ))/(EC_LE)". But in Line 204, it's defined as "((EC_LE )-(HA_LE ))/(HA_LE )". Please clarify it. | The expression for the relative difference was corrected to ((EC_LE )-(HA_LE ))/(EC_LE), it was not good in the text of the paragraph. Thanks for spotting it. |
| P11, Figure 5: Is this the monthly mean surface $NO_2$ concentrations in April 2019? Please clarify this in the caption. | Figure 5 displays an instantaneous image of the $NO_2$ columns for two model configurations, and the difference between them, during the TROPOMI overpass at the same time. An additional image was added on the same panel to show the plume structures on another day when conditions were more extreme and drove westward. A monthly mean will not show the differences in this detail, as the average over changing conditions would hide the underlying cause of the differences. |

| | |
|---|---|
| P12, Line 215-226: Why did the authors only compare the simulations with the TROPOMI observations on April 22? I would suggest the authors to do a general comparison during April 2019 and provide more accurate statistics for model validations. | The specific date of the April 22 was chosen because of the clear plumes visible in the TROPOMI data. In addition, the figures in the appendix show the wind direction and speed magnitude from the sensors in Cabauw for two episodes, with in one case a transition of wind direction. |
| P13, Figure 6: I would suggest the authors to unify the units of tropospheric column of $NO_2$ simulated by two experiments and observed by TROPOMI. Please use the same color bars and add the same map in Figure 3c. I cannot identify which experiment performed better compared to the TROPOMI observations based on current figure. | The units were unified as well as the color bars and the base map to improve the comparison. The comparison with the TROPOMI observations is on a qualitative basis, details of the evaluation of two model configurations and TROPOMI will be addressed in a follow-on paper. |
| P12, Line 235-237: "These differences may be ttributed to using different meteorological and mission data in the two configurations…". Why id the authors use different emission data in th wo cocorrectedtions? Based on Table 2, the mission data should be the same in the two xperiments. | We thankt te reviewer for spotting this mistake, we corrected it in the paragraph mentioned. The emission data is the same for the two experiments, the difference in the two systems is the meteorology. |
| P12, Line 227-237: I would suggest the authors to provide more informative discussion on the differences of $NO_2$ concentrations and Kz coefficients. I'm wondering what meteorological factors cause the differences of the simulated $NO_2$ concentrations. Why did the authors only analyze the Kz coefficients? Based on Figure 7, it's hard to say which experiment performed better in simulating $NO_2$ concentrations. Again, I'd like to suggest the authors to quantify the differences between the simulations and the observations. | In the supplementary material, an extra image with the comparison of the boundary layer height has been included to have more qualitative information on the main differences between the two meteorological drivers. In the general comments of this paper we commented on the issue concerning vertical transport. |

| Technical corrections | Reaction from the authors |
|---|---|
| P2, Line 43: "… (van Stratum et al., 2022)" should be "… van Stratum et al. (2022)". | Modified as suggested by the reviewer. |
| P4, Line 111: Please spell out the acronyms "ECGATE" to "SNELLIUS" when they first appear. | These acronyms refer to the computing servers, but have been removed from the revised manuscript as considered not relevant. |
| P9, Line 169: Please spell out the acronyms "TROPOMI" and add citations. | Acronyms are defined as suggested by the reviewer. |
| P12, Line 232: What's the Kz coefficient? | $K_z$ is the vertical diffusion coefficient |
| P14, Figure 8: Please add (a) and (b) in the upper and lower panels. | Figure 8 was labeled with the (a) and (b). |
| P15, Figure 9: Please add (d) in the panel. Please clarify the unit "[m[2] s−1]" of Kz. | The label (d) was added to the panel and the unit was corrected in the caption. |